# Isoliquiritigenin Inhibits the Growth of Colorectal Cancer Cells through the ESR2/PI3K/AKT Signalling Pathway

**DOI:** 10.3390/ph17010043

**Published:** 2023-12-27

**Authors:** Fenglin Luo, Yimeng Tang, Lin Zheng, Ying Yang, Haoyue Gao, Shiya Tian, Hongyu Chen, Chenxi Tang, Shanshan Tang, Qiong Man, Yiying Wu

**Affiliations:** 1Department of Pharmacology, School of Pharmacy, Chengdu Medical College, Chengdu 610500, China; luofenglin228@163.com (F.L.); 15881611404@163.com (Y.T.); 17394937787@163.com (L.Z.); yangyinger225@163.com (Y.Y.); 18085948114@163.com (S.T.); 13990523337@163.com (H.C.); tangchenxi0612@139.com (C.T.); m13518416103@163.com (S.T.); 2Department of Pharmacy, Study on the Structure-Specific Small Molecule Drug in Sichuan Province College Key Laboratory, Chengdu Medical College, Chengdu 610500, China; 3Department of Geriatrics, Women and Children, School of Nursing, Chengdu Medical College, Chengdu 610106, China; haoyue.gao@foxmail.com

**Keywords:** colorectal cancer, Isoliquiritigenin, network pharmacology, ESR2, PI3K/AKT pathway

## Abstract

Colorectal cancer (CRC) is one of the most common malignancies. Isoliquiritigenin (ISL), a flavonoid phytoestrogen, has shown anti-tumour activities against various cancers. However, its anti-CRC mechanism has not been clarified. In this study, the potential molecular mechanism of ISL against CRC was investigated through network pharmacological prediction and experimental validation. The results of the network prediction indicate that ESR2, PIK3CG and GSK3β might be the key targets of ISL against CRC, which was verified by molecular docking, and that its anti-tumour mechanisms might be related to the oestrogen and PI3K/AKT signalling pathway. The experimental results show that ISL reduced the viability of SW480 and HCT116 cells, induced apoptosis, blocked the cell cycle in the G2 phase in vitro, and suppressed xenograft tumour growth in vivo. In addition, ISL significantly down-regulated the protein expression of PIK3CG, AKT, p-AKT, p-GSK3β, CDK1, NF-κB and Bcl-2; up-regulated ESR2 and Bax; decreased the ratio of p-AKT/AKT and p-GSK3β/GSK3β; and increased the Bax/Bcl-2 ratio. This study indicates that ISL can inhibit the growth of CRC cells and induce apoptosis, which may be related to the up-regulation of ESR2 and inhibition of the PI3K/Akt signalling pathway.

## 1. Introduction

Colorectal cancer (CRC) is one of the most common human malignancies. According to a statistical analysis of global cancer in recent years, CRC has become the second leading cause of cancer death, with a mortality rate of up to 9.4% [1,2]. Epidemiological studies have shown that the occurrence of CRC is closely associated with gender, age, lifestyle (alcoholism, obesity, lack of physical activity, unhealthy diet, etc.), living environment and family history [3]. Currently, the primary therapeutic methods for CRC include surgery, postoperative chemotherapy, chemotherapy and neoadjuvant radiotherapy, targeted therapy, maintenance therapy and immunotherapy [4,5,6,7,8,9].

However, surgical injury, postoperative recurrence and metastasis, and toxic side effects of chemoradiotherapy, such as leukopenia, thrombocytopenia and severe diarrhoea, not only affect the overall therapeutic effect but also seriously reduce the quality of life of patients [10]. Recently, botanical drugs with anti-cancer activity have attracted much attention due to their low toxicity and efficacy [11]. Therefore, the search for natural products to prevent and treat CRC has become a hotspot in the field of cancer research [12].

Several studies have found that oestrogen might play a potential role in the prevention and treatment of CRC [13,14,15]. Phytoestrogens are a class of non-steroidal plant compounds which could simulate the characteristics of oestrogen in vivo [16]. Isoliquiritigenin (2′,4′,4-trihydroxy chalcone; ISL), a natural chalcone compound, is one of the most extensively studied phytoestrogens [17]. It has been found to have a broad range of pharmacological effects, such as anti-inflammatory, anti-oxidant, anti-allergic, anti-cancer, anti-arrhythmia, liver protection and immunomodulatory effects [18]. ISL is extracted from liquorice root, a commonly used traditional Chinese herb and an excellent food plant resource. Therefore, liquorice is considered as a potential medicinal resource, and the pharmacological effects of its main active ingredients have been extensively studied. ISL is one of the main active ingredients in liquorice. Relevant studies have shown that ISL can effectively inhibit the growth and metastasis of various malignant tumours such as breast cancer, ovarian cancer, lung cancer and gastric cancer, and its anti-tumour effect in CRC has gradually begun to be identified. Still, the mechanism of its anti-CRC effect remains unclear [19].

Network pharmacology is a novel strategy for predicting drug active ingredients and disease targets based on computer science and system biology. By constructing a drug-ingredient-target-disease biological system network, the interrelationship between active ingredients and disease targets is revealed [20]. And NP is now widely used in the screening of active ingredients and mechanism exploration in traditional Chinese medicine [21,22].

In this study, we attempted to investigate the anti-CRC effect and mechanism of ISL. First, we predicted the key targets and possible mechanisms of ISL against CRC through network pharmacological analysis. Then, we adopted molecular docking and pharmacological experiments in vitro and in vivo to verify the anti-CRC effects and molecular mechanisms of ISL. A comprehensive approach based on network pharmacological prediction was utilised for the first time to explore the molecular mechanism of ISL against CRC. This study provides a candidate compound for the treatment of CRC and a strategy for studying the molecular mechanisms of natural products.

## 2. Results

### 2.1. Target Prediction of ISL for Prevention and Treatment of CRC

A total of 31 corresponding targets of ISL and 5473 disease genes associated with CRC were screened out from the database. Twenty-seven common drug–disease targets were obtained from the intersection between CRC disease-related gene targets and ISL pharmacological targets, as shown in Figure 1A. The PPI network diagram of common drug–disease targets was constructed using the STRING database (Figure 1B). As shown in the ISL-target-CRC relationship network diagram (Figure 1C), 29 nodes, 137 edges and an average degree value of 9.45 were obtained.

To delve deeper into the mechanism of ISL against CRC, we conducted GO (Gene Ontology) gene enrichment analysis and KEGG (Kyoto Encyclopedia of Genes and Genomes) pathway analysis for these common targets. According to the GO enrichment analysis (Figure 1D), the effect of ISL on CRC not only involves the regulation of biological processes such as nitric oxide biosynthesis, protein entry into the nucleus, transcriptional regulation and DNA templating, but it also involves the regulation of cellular components such as the nucleus and cytoplasm. It is also associated with protein binding, RNA polymerase II transcription factor activity and the activation of ligand sequence-specific DNA binding. KEGG enrichment analysis (Figure 1E) revealed that the key targets of ISL that antagonise CRC were mainly concentrated in the oestrogen signalling pathway, TNF (tumour-necrosis-factor) signalling pathway, PI3K/AKT (phosphatidylinositol-3-kinase/protein kinase) signalling pathway and other classical signalling pathways. Among them, the PI3K/AKT signalling pathway has been recognised as being closely associated with the development of CRC [23,24,25]. In addition, the anti-tumour effect of ISL, as a typical phytoestrogen, has been found to be related to oestrogen receptors [26,27,28]. It has been found that oestrogen receptors can participate in tumour growth and metastasis by regulating the PI3K signalling pathway [29]. ESR2 (oestrogen receptor beta), PIK3CG (phosphatidylinositol-4,5-bisphosphate 3-kinase catalytic subunit gamma) and GSK3β (glycogen synthase kinase three beta), the predicted common drug–disease targets, are typical targets involved in the oestrogen signalling pathway and the PI3K/AKT pathway. Therefore, they are considered to be key targets for ISL intervention in CRC.

### 2.2. Molecular Docking Validation of the Predicted Key Targets

To confirm the accuracy of the above prediction, ISL was molecularly docked with ESR2, PIK3CG and GSK3β proteins. In the formation of conformation stability, the lower the energy when the drug molecule ligand binds to the protein receptor, the more stable the structure, the better the fit and the more effective it can be. ISL exhibited a high binding affinity with ESR2, PIK3CG and GSK3β, with binding energies of −8.2, −7.9 and −7.6 kcal/mol, respectively. This suggests that ISL has a high binding affinity for predicting key targets. As shown in Figure 2, ISL can interact with ESR2 through the formation of two hydrogen bonds with LEU339 and ARG346. It also interacts with PI3KCG by forming three hydrogen bonds with LYS1045, SER594 and PRO590, and interacts with GSK3β by forming two hydrogen bonds with LYS85 and CYS199. These results suggest that ISL may have a therapeutic role in treating CRC due to its synergistic effects on multiple targets.

### 2.3. ISL Inhibits CRC Cell Viability, Induces Cell Apoptosis and Blocks Cell Cycle

The results of the MTT experiment are presented in Figure 3A. Compared with the negative control group, the OD value at 490 nm gradually decreased (*p* < 0.001) with the increase in ISL concentration. At a concentration of 140 μmol/L, the inhibitory rate of ISL against SW480 cells was 74.12 ± 0.039%, and its inhibitory rate against HCT116 cells was 82.67 ± 0.018%. The results indicate that ISL suppressed the viability of SW480 and HCT116 cells in a dose-dependent manner.

To further determine whether ISL induced apoptosis of CRC cells, flow cytometry was employed to assess the levels of apoptosis in cells subjected to ISL treatment. As depicted in Figure 3B, compared with the control group, the apoptosis in SW480 and HCT116 cells could be significantly increased after treatment with ISL for 24 h (*p* < 0.001). The total apoptosis rates of SW480 and HCT116 cells treated with 80 μmol/L ISL were 43.21 ± 4.84% and 61.36 ± 5.37%, respectively. Additionally, following ISL treatment, the population of G0/G1-phase cells in HCT116 and SW480 cells were reduced, whereas the proportion of cells in the S and G2/M phases increased. These findings indicate that ISL can reduce the viability of CRC cells, induce apoptosis and block cell cycle progression in the G2/M phase.

### 2.4. ISL Up-Regulates ESR2 and Inhibits the PI3K/AKT Signalling Pathway in CRC Cells

Next, to validate the predicted molecular mechanism of ISL against CRC using network pharmacology, the expression of ESR2- and PI3K/AKT-signalling-pathway-related proteins was evaluated. Indeed, Western blot analysis revealed that ISL significantly up-regulated the protein expression of ESR2 and Bax (*p* < 0.05) and down-regulated the expression levels of PIK3CG, AKT, P-AKT, P-GSK3β, CDK1, NF-κB and Bcl-2 (*p* < 0.05) in SW480 and HCT116 cells. Furthermore, ISL decreased the P-AKT/AKT and P-GSK3β/GSK3β ratios (*p* < 0.05) and increased the Bax/Bcl-2 ratio (Figure 4A). Meanwhile, the immunofluorescence staining results of intracellular protein expression changes after ISL treatment were consistent with the Western blot results, as shown in Figure 4B. This suggests that ISL may inhibit the PI3K/AKT signalling pathway by up-regulating ESR2, thereby suppressing the growth of CRC cells.

### 2.5. ISL Induces ROS Accumulation in CRC Cells

ROS is a class of compound reactive chemicals containing oxygen. Studies have shown that a low concentration of ROS acts as a second messenger to mediate a variety of signal transduction pathways in cells, playing a role in regulating cell protection and promoting cell proliferation. However, a high concentration of ROS promotes the occurrence of oxidative stress and induces autophagy and apoptosis [30]. In this study, flow cytometry was employed to assess alterations in ROS levels within CRC cells following a 24-h treatment with ISL. The results indicate that ISL significantly increased ROS levels in both HCT116 and SW480 cells (Figure 5A). Indeed, fluorescence microscopy indicated a significant increase in the expression of ROS after ISL treatment compared to the control group (Figure 5B). The above experimental results indicate that ISL may promote ROS accumulation in CRC cells.

### 2.6. ISL Inhibits the Tumour Growth In Vivo

To further investigate the anti-CRC effects of ISL in vivo, a mouse-transplanted tumour model of CRC was established. The results demonstrate that there were no significant differences in body weight between mice in the ISL treatment group and those in the control group (Figure 6A). The transplanted tumours exhibited decreased growth rates and weights in the ISL treatment group (*p* < 0.01) (Figure 6B). The tumour inhibition rates in the ISL 30 mg/Kg group and the ISL 60 mg/Kg group were 28.13% and 35.11%, respectively (Figure 6C).

Immunohistochemical analysis revealed that the protein expressions of PI3KCG, AKT and P-GSK3β in tumour tissues of ISL-treated mice were down-regulated, while the expression of ESR2 was significantly elevated (*p* < 0.05) (Figure 6D). In addition, an immunofluorescence assay was employed to detect the apoptotic gene expression in the xenograft tumour tissues of mice, and the results show the expression of Cleaved Caspase-3 in the ISL treatment group was significantly increased (*p* < 0.05) (Figure 6E). Consistent with the results of the in vitro cell experiments, it was found that ISL can suppress the growth of colorectal tumours and regulate the ESR2 and PI3K/AKT pathways in vivo.

## 3. Discussion

ISL is the main active ingredient of liquorice, a common traditional Chinese herb in China. As a typical phytoestrogen, ISL has shown significant anti-tumour effects in various sex-hormone-dependent cancers, such as breast, ovarian and prostate cancer [31,32,33]. Recently, several studies have demonstrated the inhibitory effect of ISL on non-sex-hormone-dependent tumours [34]. For example, Tian et al. found that ISL exerts inhibitory effects on the invasion and migration of non-small-cell lung cancer A549 cells through the modulation of the PI3K/AKT signalling pathway [35]. In CRC tumour cells, ISL has been found to inhibit cell growth and induce apoptosis [36]. This study further confirmed the inhibitory effects of ISL on CRC cells both in vivo and in vitro and studied its molecular mechanism.

Network pharmacology is one of the research strategies for developing new drugs as it helps to expand the space of available targets for drugs. It has been widely used in the research of new medicines, the prediction of mechanisms of drug action and the prediction of disease therapeutic targets [37,38,39]. Our study predicted the targets of ISL for the prevention and treatment of CRC through network pharmacological analysis. Further, we verified its molecular mechanism using molecular docking and in vivo and in vitro pharmacological experiments. This research is the first to explore the anti-CRC mechanism of ISL using a comprehensive strategy of network pharmacological prediction and pharmacological experiment validation. We found that ISL may inhibit CRC tumour growth by regulating the oestrogen receptor and PI3K/AKT signalling pathway (Figure 7).

The oestrogen receptor (ER) is a crucial target protein that affects tumorigenesis and progression. It can be divided into two subtypes, ERα (ESR1) and ERβ (ESR2), depending on the transcription gene [40,41]. It was found that ESR2 is the primary ER expressed in the colonic epithelium, and its expression is reduced during colon tumorigenesis [14]. The up-regulation of ESR2 can inhibit the occurrence and progression of CRC [15]. PI3K/Akt is one of the crucial signalling pathways in cell survival, playing an essential role in regulating a variety of cellular functions [42]. It is related to regulating tumour cell proliferation, apoptosis, invasion, metastasis, drug resistance and the immune microenvironment [43]. PI3K is a large family of enzymes in which a catalytic subunit (P110) converts PIP2 to PIP3, activating downstream kinases such as Akt. Akt plays a crucial role in regulating various downstream effector molecules, such as GSK3β, mTOR, NF-κB, Bcl-2 and other effector molecules through the phosphorylation cascade, thereby controlling cell biological processes [44,45,46]. A previous study showed that the overexpression of ESR2 can increase the PTEN protein, thereby inhibiting the PI3K/Akt signalling pathway and inhibiting breast cancer cells [47]. Tian et al. found that apoptosis of osteosarcoma cells can be induced by ESR2-dependent inhibition of the PI3K/Akt signalling pathway [48]. In addition, Zhu et al. found that calycosin, as a phytoestrogen, can inhibit the growth of CRC by targeting ESR2 and inhibiting the PI3K/Akt pathway [29]. These studies suggest that the up-regulation of ESR2 to suppress the PI3K/AKT signalling pathway might be a novel strategy for preventing and treating CRC.

Based on the predicted results obtained through network pharmacology, this study detected the expression of ESR2- and PI3K/AKT-pathway-related target proteins in CRC cells after ISL treatment. The results show that ISL significantly decreased the protein expression of PIK3CG, p-AKT, AKT, NF-κB, p-GSK3β and CDK1 in CRC cells, while it enhanced the expression of ESR2. Additionally, the ratios of p-AKT/AKT and p-GSK3β/GSK3β exhibited a dose-dependent decrease after ISL treatment. The BCL-2 protein family, including the pro-apoptotic gene Bcl2 and the anti-apoptotic gene Bax [49], plays a crucial role in regulating intrinsic apoptosis. In this study, ISL remarkably increased the expression of the Bax protein and down-regulated the Bcl2 protein expression, while it increased the Bax/Bcl2 ratio in CRC cells. These results confirm the network pharmacological prediction that ISL can inhibit CRC through the ESR2/PI3K/AKT signalling pathway.

This study also revealed that ISL can increase intracellular ROS accumulation in CRC cells, which is closely related to ferroptosis. Ferroptosis, a newly discovered non-programmed cell death process, is characterised as being iron-dependent and peroxidation-driven, resulting in lethal lipid ROS [50,51]. Previous studies have shown that activating ferroptosis via the promotion of ROS production in cells can effectively inhibit the occurrence and progression of CRC [52,53]. Therefore, this study proposes that the anti-CRC effect of ISL may also be linked to the intracellular accumulation of ROS, thus inducing cell ferroptosis, which is worthy of in-depth research.

In addition, the TNF signalling pathway may also be involved in the anti-CRC effect of ISL, which is predicted by network pharmacology. TNF is a major mediator of apoptosis, inflammation and immunity and is involved in the pathogenesis of a variety of human diseases, including cancer. It has been demonstrated that the TNF pathway may take part in the initiation and progression of CRC through the activation of MAPK, JNK/AP-1 and NF-κB pathways, thereby enhancing oncogene activation, promoting pro-inflammatory cytokine release, increasing colonic epithelial cell proliferation, and form a tumour-supportive Tumour Microenvironment [54]. In addition, tumour necrosis factor (TNF)-related apoptosis inducing ligand (TRAIL), a member of the TNF superfamily, has been implicated in cancer treatment. It interacts with death receptors (DRs) and induces apoptosis of various tumour cells. However, some cancer cells are resistant to TRAIL-induced apoptosis, such as colorectal tumours. Numerous studies have shown that natural products can enhance the susceptibility of CRC cells to TRAIL-induced apoptosis, suggesting that natural TRAIL-sensitising agents can help treat tumours [55]. ISL has also been found to sensitise TRAIL. It can up-regulate DR5 protein, one of the TRAIL receptors, which enhances the apoptosis-inducing activity of TRAIL in CRC cells HT29 [56]. These findings suggest that the regulation of TNF and TRAIL pathways may also be the anti-CRC mechanisms of ISL. However, these were not explored in this study and may be considered for further investigation.

## 4. Materials and Methods

### 4.1. Prediction of Targets of ISL against CRC

The pharmacological targets of ISL were predicted using the TCMSP database (https://old.tcmsp-e.com/tcmsp.php, accessed on 20 May 2022), resulting in the identification of 31 targets. Meanwhile, 5473 therapeutic targets related to CRC were collected from the DisGeNET database (http://www.disgenet.org/, accessed on 20 May 2022) [57]. Then, 27 common targets were determined by comparing the pharmacological targets of ISL with the targets related to CRC. The protein interaction relationships among these common targets were established using the STRING database (https://www.string-db.org/, accessed on 22 May 2022). The ISL-target-CRC network was constructed and visualised using Cytoscape databases. In addition, bioinformatics analyses of 27 common targets were performed using DAVID Bioinformatics Resource 6.8 software, which included GO analysis and KEGG pathway enrichment analysis.

### 4.2. Molecular Docking

To verify the accuracy of the network pharmacology prediction results, ISL was docked with the key targets including ESR2, PIK3CG and GSK3β. The 3D structures of active constituents of ISL and critical targets (https://www.rcsb.org/, accessed on 25 May 2022) were collected. The proteins and ligands were segmented using pymol software (1. 8. X), and then the water molecules and excess amino acid sequences of the proteins were removed. AutoDuckTools-1.5.6 was used to convert the processed proteins, drugs and ligands into corresponding formats. Finally, the AutoDock Vina program was used to perform the molecular docking analysis [58]. The crystal structures utilised for ESR2, PIK3CG and GSK3β, along with the parameter settings for the grid box, are provided in Appendix A Appendix A.

### 4.3. Treatment of ISL

ISL was purchased from Dalian Meilun Biotechnology Co. (Dalian, China), LTD. (MB2209-S, purity > 99%). The 200 μmol/L ISL solution was dissolved in DMSO and stored at −20 °C, and then it was diluted to the final concentration with DMEM or RPMI-1640 medium.

### 4.4. Culture of CRC Cell SW480 and HCT116

Human CRC HCT116 (CL-0096), human SW480 (CL-0223A) and mouse CT-26 (CL-0071) cells were purchased from Procell and cultured in an incubator with 5% CO_2_ and at a temperature of 37 °C. The composition of the culture medium consisted of DMEM or RPMI-1640/newborn bovine serum/double antibody (penicillin + streptomycin) = 10:1:1.

### 4.5. Cell Viability Assay

The SW480 and HCT116 cells in the logarithmic growth phase were seeded into a sterile 96-well plate at a density of 1 × 10^4^ cells per well. Then, 24 h after inoculation, the SW480 cells were treated with 0, 20, 60, 100, 140, 180 and 220 μmol/L of ISL, and the HCT116 cells were subjected to treatment with different concentrations of ISL (0, 20, 40, 60, 80, 100 and 120 μmol/L), with five re-wells for each concentration. After 48 h of treatment, 10 μL of 3-(4,5-dimethyl-2-thiazolyl)−2,5-diphenyl-2-H-tetrazolium bromide (MTT) reagent was added to each well. After incubating again for 4 h and removing the medium, DMSO solvent was added to each well. The optical density (OD) value of each well at the 490 nm wavelength was measured using an enzyme-labelled instrument (BioTek, Winooski, VT, USA). Cell survival and inhibition rates were calculated according to the following formulas: cell survival rate (%) = (experimental group A490 nm − blank group A490 nm)/(cell control group A490 nm − blank group A490 nm) × 100%; cell inhibition rate (%) = 1 − cell survival rate (%). The experiment was repeated 3–5 times.

### 4.6. Apoptosis and Cell Cycle Assays

After treatment with ISL for 24 h, the cells were collected and washed twice with cold PBS. Next, 5 μL of Annexin V–FITC (TRANSGEN), 5 μL of propidium iodide (PI) (TRANSGEN) and 50 μL of 1× binding buffer were added to each tube. The solution was then gently mixed and incubated at room temperature for 15 min in the dark. Then, 200 μL of 1× binding buffer was added to each tube. In addition, the cells were treated with 75% ethanol at 4 °C for 12–24 h, followed by staining with PI (BD Biosciences, Franklin Lakes, NJ, USA) for the cell cycle detection. Finally, flow cytometry (ACEA NovoCyte, Santa Clara, CA, USA) was used to explore cell apoptosis and cycle distribution [59].

### 4.7. Western Blot

The CRC SW480 and HCT116 cells were lysed with cell lysate, and the concentrations of extracted protein were detected using a BCA protein quantification kit (CW0014S, cwbio). Then, the protein was denatured in a metal bath at 100 °C for 10 min. The protein samples were packed into 12% SDS-PAGE for electrophoresis separation. Finally, a 0.22 μm PVDF membrane was used as the transmembrane. After membrane transfer, the PVDF membrane was sealed with 5% BSA sealing solution for about 2 h. The following primary antibodies were added and incubated overnight at 4 °C: anti-AKT (Proteintech (Rosemont, IL, USA), 1:2000), anti-p-AKT (Abways (Shanghai, China), 1:2000), anti-ESR2 (Proteintech, 1:2000), anti-CDK1 (Proteintech, 1:2000), anti-p-GSK3B (Proteintech, 1:2000), anti-NF-ĸB p65 (Proteintech, 1:2000), β-actin (Proteintech, 1:5000), anti-Bcl2 (Proteintech, 1:3000), anti-Bax (Proteintech, 1:5000), anti-GSK3B (Proteintech, 1:5000) and anti-PI3K gamma (Bioss, 1:1000). Subsequently, the primary antibody was recovered, and the secondary antibody was incubated for one hour at room temperature. The membranes were washed 3 times with TBST (5 min each time), the blot was visualised using the chemiluminescence method. The band intensities were quantified using ImageJ-win64.

### 4.8. Immunofluorescence

The SW480 and HCT116 cells were inoculated into a sterilised 6-well plate at 2 × 10^5^ cells per well. After being left overnight, the cells were treated with ISL with concentration gradients of 0 and 80 μmol/L for 24 h. The cells were rinsed three times with pre-cooled PBS, fixed with 4% paraformaldehyde for 30 min at room temperature and then permeabilised with 5% triton-100 for 20 min at 4 °C. A sealing solution was added, and it was incubated for one hour at 37 °C. The primary antibody (1:200) was added and left to incubate overnight at 4 °C. After washing with pre-cooled PBS 3 times (10 min each), the fluorescent secondary antibody (1:50) was added and left to incubate at 37 °C for 2 h. The cell nucleus was stained with DAPI (4′,6-diamidino-2-phenylindole) for 10 min. Finally, the slide was sealed using an anti-fluorescence quencher and subjected to observation by an Olympus BX63 fluorescence microscope (Olympus, Tokyo, Japan) at a magnification of 400×.

### 4.9. ROS Detection

The SW480 and HCT116 cells were inoculated into a six-well plate and treated with or without ISL for 24 h. Cells were collected or in situ loaded with a DCFH-DA fluorescent probe (diluted in serum-free medium at 1:1000) and incubated in an incubator at 37 °C for 20 min. Following that, the cells were washed three times using a serum-free culture solution to remove the probe. The intracellular reactive oxygen species (ROS) fluorescence intensity of the cells was detected using flow cytometry or inverted fluorescence microscopy.

### 4.10. Animals and Treatment

Twenty-four SPF BALB/c male mice (SYXK (Chuan)2020-196), weighing 18–20 g, were purchased from Dossy In Vitro Animal Co., Ltd. (SCXK (Chuan) 2020-0030, Chengdu, China), and were kept under standard laboratory conditions of a temperature of 23 ± 1 °C, humidity of 55 ± 5% and a 12-h light/dark cycle. Following the recommendations of the Medical Ethics Committee of Chengdu Medical College and the Declaration of Helsinki, a mouse xenograft tumour model was established. In total, 1 × 10^7^ CT-26 cells were implanted into the flanks of the mice. When the tumour volume reached the standard (40–60 mm^3^), 18 mice with a relatively uniform body weight and tumour size were divided into three groups: the ISL 30 mg/Kg group, the ISL 60 mg/Kg group and the saline solution control group. Mice in each group were intraperitoneally injected with ISL or saline for 14 days. The weight and tumour volume of the mice were observed and recorded daily. The tumour volume (mm^3^) was calculated according to the following standard formula: V = Length × Width^2^/2. Finally, the mice were euthanised, and tumour specimens were collected.

### 4.11. Immunohistochemistry and Immunofluorescence Experiments

The paraffin tumour tissue was continuously sliced at 4 μm and roasted at 60 °C for 2 h. The slices were then dewaxed with xylene and hydrated with a gradient of gradient ethanol. In order to repair antigens, paraffin sections were immersed in sodium citrate buffer and heated using a microwave oven twice for 2–3 min. Then, 5% bovine serum albumin (BSA) was added, and the sections were incubated at 37 °C for 30 min to block non-specific signals after washing with PBS twice for 5 min. The blocking solution was then removed, and ESR2, PIK3CG, P-AKT, P-GSK3β and Cleaved Caspase-3 antibodies (dilution 1:200) were added to the sections and then left to incubate overnight at 4 °C. After washing with PBS for 3 min, the slices were incubated at room temperature in the dark for 1 h with horseradish peroxidase (HRP) [60] or Alexa Fluor-secondary antibodies (1:1000, Proteintech Group, Inc., Rosemont, IL, USA). Then, diaminobenzidine (DAB) was added to the slides for 3–10 min to develop colour, or DAPI was added for 10 min to stain the nuclei. Finally, the slices were sealed with a neutral gum or anti-fluorescent quencher. The staining was observed and photographed with an Olympus BX63 fluorescence microscope (Olympus, Tokyo, Japan) at 400×. The staining results were evaluated using the ImageJ analysis software (win64), and the mean optical density (MOD) was used to quantify the protein expression (MOD = combined selection density/area).

### 4.12. Statistical Analysis

Statistical analyses were performed using GraphPad Prism 8.0 and SPSS21.0. Results are shown as the mean ± standard deviation (SD), and differences between groups were analysed using one-way ANOVA. *p*-values of <0.05 were considered significant.

## 5. Conclusions

In conclusion, this study is the first to explore the mechanism of ISL against CRC by using a comprehensive strategy of network pharmacological prediction and in vitro and in vivo pharmacological experiment validation. It was found for the first time that ISL may inhibit the PI3K/AKT signalling pathway by targeting ESR2, thereby inducing apoptosis and inhibiting the growth of CRC cells. It is suggested that ISL has the potential to be a novel therapeutic agent for CRC, and ESR2 may be the anti-tumour target of ISL and other phytoestrogens. However, the study on the anti-CRC effect of ISL is not deep enough. In order to further explore the molecular mechanism of ISL against CRC, inhibitors or gene silencing of ESR2 will be considered in future studies. In addition, the anti-CRC effect of ISL may also be related to ferroptosis, TNF-mediated inflammation and apoptosis, which needs more in-depth studies to explore.

## Figures and Tables

**Figure 1 pharmaceuticals-17-00043-f001:**
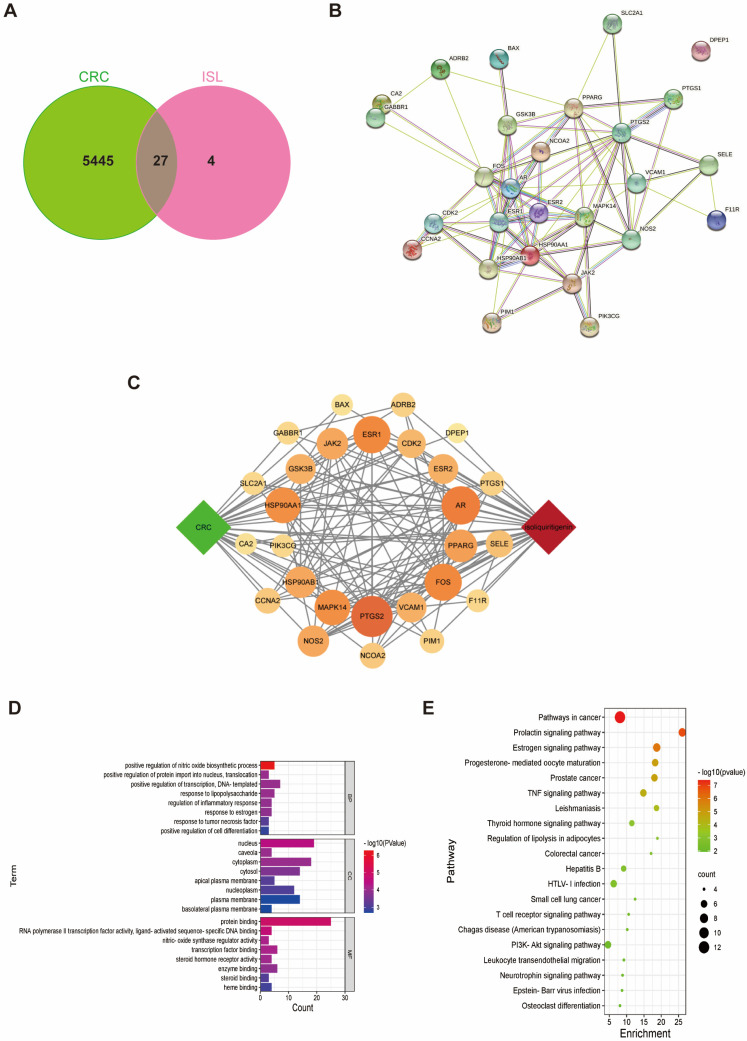
Network Pharmacology Analysis of anti-CRC mechanism of ISL. (**A**) Venn diagram of the overlapping target genes of ISL and CRC. (**B**) The PPI network of common targets. (**C**) ISL-target-CRC network. The green node represents CRC, the orange nodes represent common targets, and the red node represents ISL. Larger node sizes and darker colours indicate a higher degree of correlation. (**D**) GO functional annotation of potential targets of ISL against CRC, including biological processes (BP), cellular components (CC) and molecular functions (MF). (**E**) KEGG enrichment analysis for potential signalling pathways. The top 20 pathways with lower *p*-values are visualised.

**Figure 2 pharmaceuticals-17-00043-f002:**
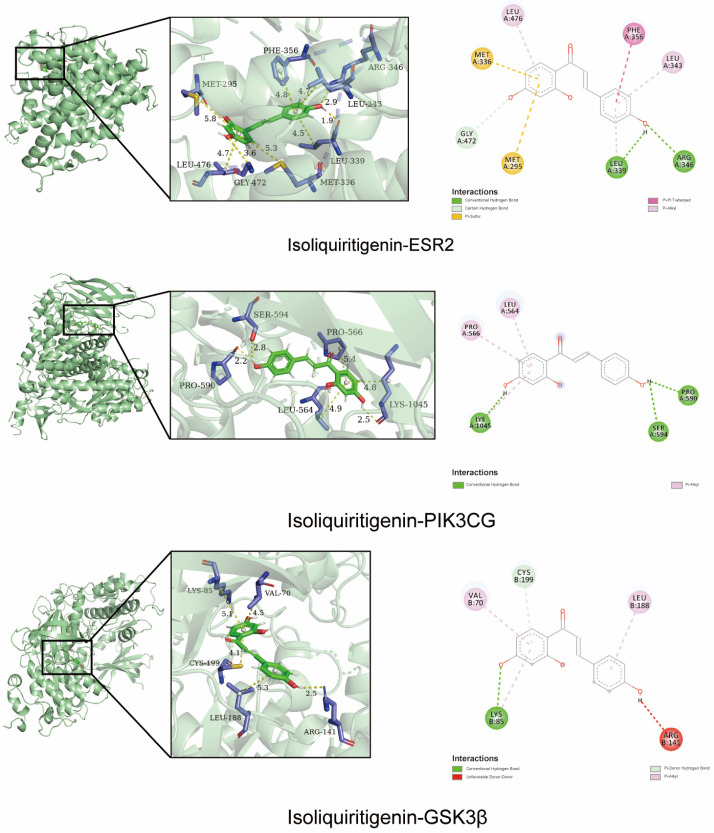
Molecular docking of ISL with ESR2, PI3KCG and GSK3β.

**Figure 3 pharmaceuticals-17-00043-f003:**
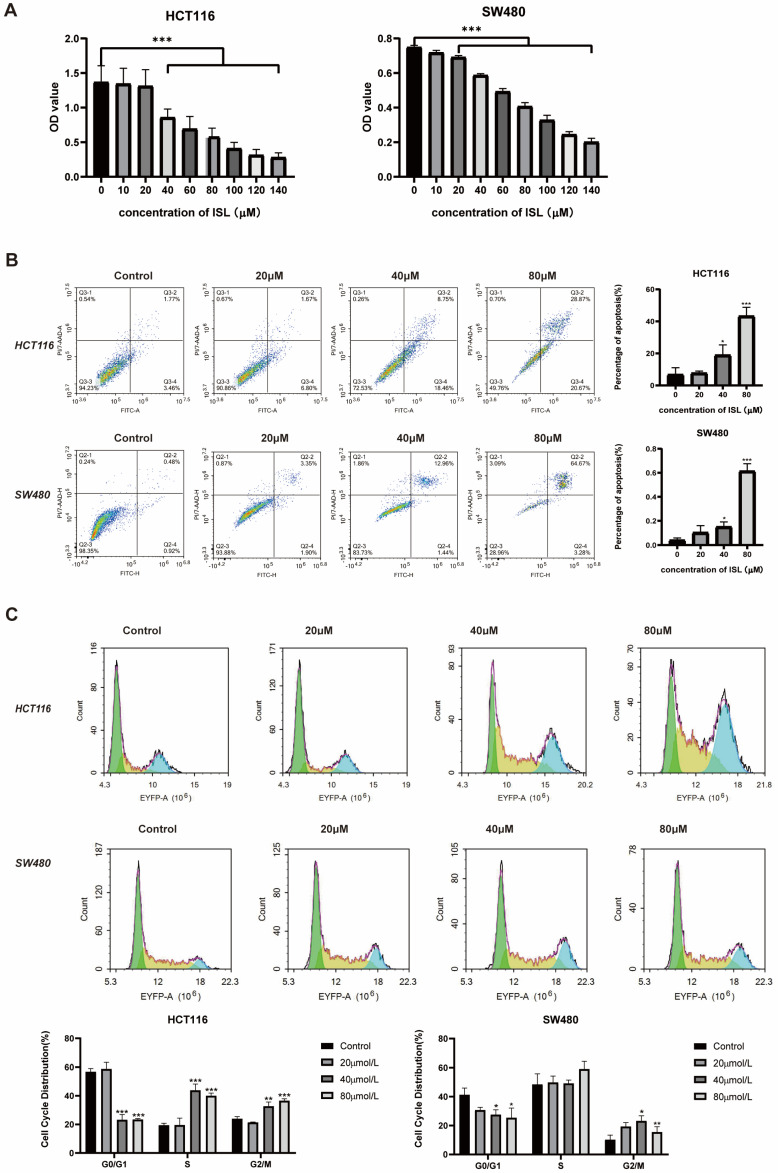
The effects of ISL on CRC cell viability, cell apoptosis and cell cycle. (**A**) ISL decreases the viability of CRC cells. Following a 24 h treatment with various concentrations of ISL, the viability of HCT116 and SW480 cells was assessed using the MTT assay. (**B**) ISL triggers apoptosis in CRC cells. Flow cytometry was utilised to assess the level of apoptotic cells in HCT116 and SW480 cells after treatment with ISL. The resulting histogram displays the percentage of cells undergoing apoptosis. (**C**) ISL blocks CRC cell cycle in G2/M phase. * *p* < 0.05, ** *p* < 0.01, *** *p* < 0.001 vs. negative group.

**Figure 4 pharmaceuticals-17-00043-f004:**
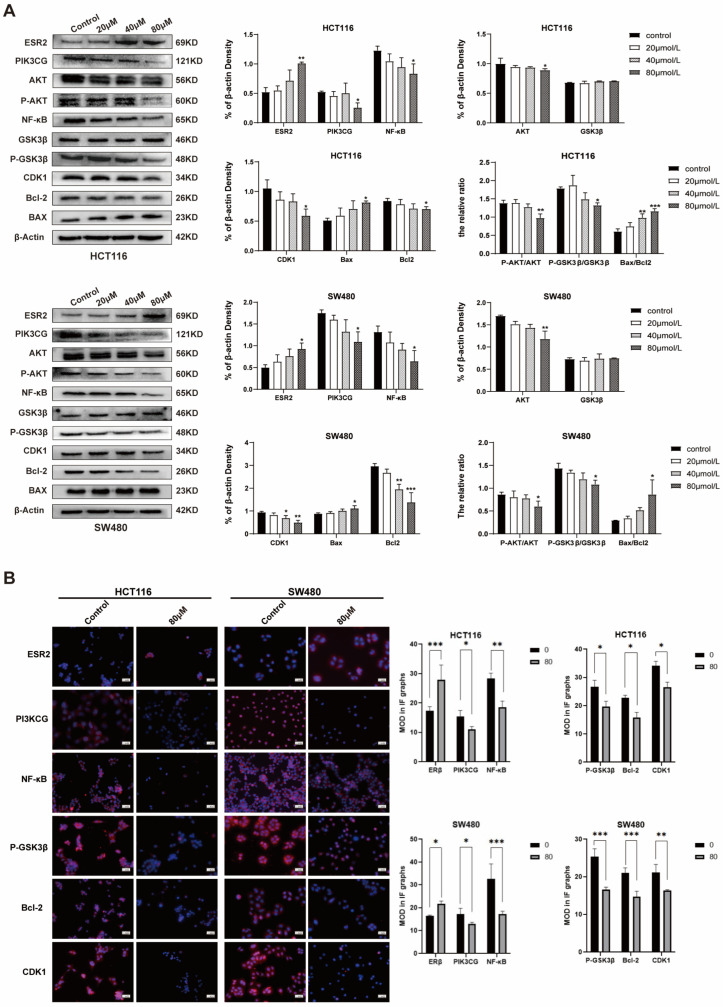
Regulation of ISL on ESR2 and PI3K/Akt signalling pathway in CRC cells. (**A**) Western blot analysis of the protein expression of ESR2, PI3KCG, NF-κB, AKT, P-AKT, CDK1, GSK3β, p-GSK3β, Bax and Bcl-2 in CRC cells. (**B**) The immunofluorescence staining results of protein expression of ESR2, PI3KCG, NF-κB, p-GSK3β, Bcl-2 and CDK1 in CRC cells. After treatment with various concentrations of ISL (0, 20, 40, and 80 µM) for 24 h, HCT116 and SW480 cells were collected to detect protein expression, with β-actin as endogenous control. * *p* < 0.05, ** *p* < 0.01, *** *p* <0.001 vs. negative control group.

**Figure 5 pharmaceuticals-17-00043-f005:**
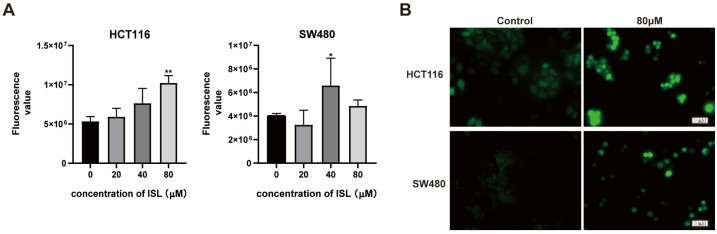
Effect of ISL on ROS accumulation in CRC cells. HCT116 and SW480 cells were treated with ISL for 24 h and examined the accumulation of ROS by flow cytometry (**A**) and immunofluorescence (**B**) assay. * *p* < 0.05, ** *p* < 0.01 vs. control.

**Figure 6 pharmaceuticals-17-00043-f006:**
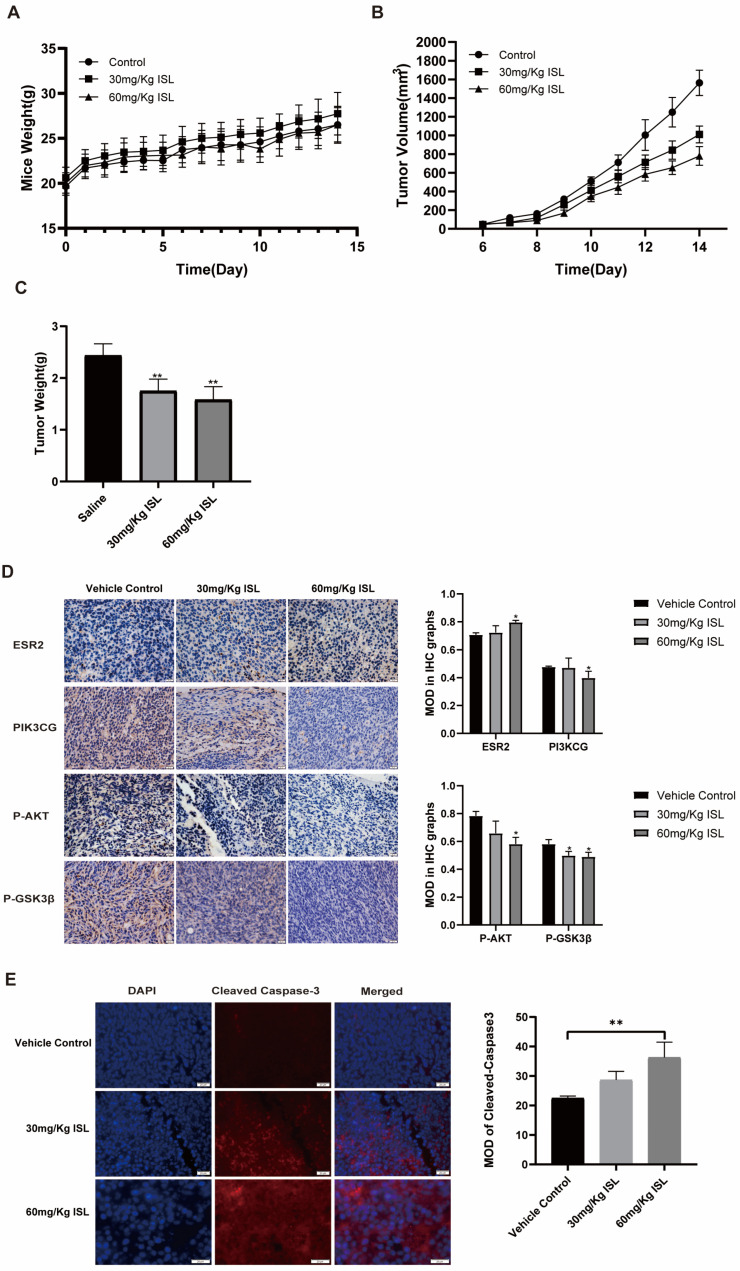
Effect of ISL on the growth of the xenograft tumour in mice. (**A**) Changes in body weight of mice. (**B**) Changes in tumour volume. (**C**) Tumour weight. (**D**) Immunohistochemistry analysis for ESR2, PI3KCG, P-AKT and P-GSK3β in xenograft tumours. (**E**) Immunofluorescence analysis for Cleaved-Caspase 3 in xenograft tumours. MOD is the mean optical density. All images were captured and displayed at a magnification of 400×. * *p* < 0.05 and ** *p* < 0.01 vs. control.

**Figure 7 pharmaceuticals-17-00043-f007:**
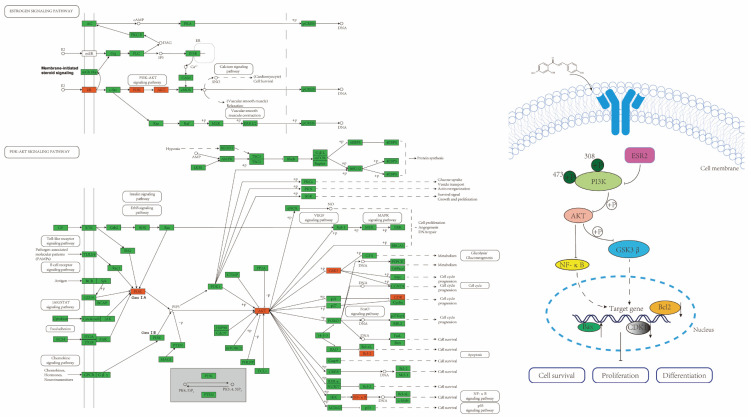
ESR2/PI3K/AKT signalling pathway.

## Data Availability

Data is contained within the article and Appendix A.

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
