# Peer review of "Isoliquiritigenin Inhibits the Growth of Colorectal Cancer Cells through the ESR2/PI3K/AKT Signalling Pathway"

_pharmaceuticals, 2023, doi:10.3390/ph17010043_

Round 1

Reviewer 1 Report

Comments and Suggestions for Authors

Manuscript ID: pharmaceuticals-2702681

The Manuscript " Isoliquiritigenin inhibits the growth of colorectal cancer cells through the ESR2/PI3K/AKT signalling pathway" by Qiong Man and Yi-ying Wu et al

The authors disclosed their recent results on the study of anti-Colorectal cancer effect of ISL and its mechanism, they also predicted the key targets and possible mechanisms of Isoliquiritigenin against Colorectal cancer through network pharmacological analysis.

Authors have carefully drafted their findings with all supporting information and provided sufficient citation in the manuscript.

However, Authors should consider providing source and or analytical supporting information for Isoliquiritigenin used in this study.

Authors can consider improving image quality for used in figure-7

This manuscript can be published, and it will be useful for scientific readers' point of view.

Author Response

Dear reviewer,

Thanks for your comments concerning our manuscript entitled “Isoliquiritigenin inhibits the growth of colorectal cancer cells through the ESR2/PI3K/AKT signalling pathway” (Manuscript ID: pharmaceuticals-2702681). Those comments are all valuable and very helpful for revising and improving our paper, as well as the important guiding significance to our researches. We have studied comments carefully and have made correction which we hope meet with approval. And the revised portion is marked in red.

1.The comment: However, authors should consider providing source and or analytical supporting for Isoliquiritigenin used in this study.

The answer:Thanks for your suggestion. We have added it in the article, the detail as follows: ISL was purchased from Dalian Meilun Biotechnology Co., LTD. (MB2209-S, purity >99%).

2.The comment: Authors can consider improving image quality for used in figure 7.

The answer: Thank you for the suggestion, which is very helpful for improving our manuscript. We have improved the quality of figure 7. It is in tiff format and 300 dpi.

Thank you very much for your attention and time, look forward to hearing from you!

Yours sincerely,

Feng-lin Luo

30, Nov, 2023

Reviewer 2 Report

Comments and Suggestions for Authors

This paper reported that isogliquiritin can inhibit the growth of colorectal cancer cells and induce apoptosis, which may be related to the up-regulation of ESR2 and the inhibition of PI3K/Akt signaling pathway, which has certain significance. However, there are many points in the paper that need to be modified and improved, such as the manufacturers and batch numbers of the main instruments provided for research; Provide the strain, specification and qualification certificate number of mice used in the test; The licence number of the animal centre for the supply of mice; There should be references in the method; There should be model group, positive control group, and the best three dose groups in the animal experiment. How to make mold should have a detailed method and mold detection index; The content of animal experiments in this study is small, with 6 mice in each group, which is a little small. The discussion part should discuss what this study reveals, the existing problems and further research in the future. Some content can be appropriately added to the conclusion; The references should be mainly from the last 3 years, and the format should be uniform. In view of the above problems, it is suggested to revise and improve the trial.

Author Response

Dear reviewer,

Thanks for your comments concerning our manuscript entitled “Isoliquiritigenin inhibits the growth of colorectal cancer cells through the ESR2/PI3K/AKT signalling pathway” (Manuscript ID: pharmaceuticals-2702681). Those comments are all valuable and very helpful for revising and improving our paper, as well as the important guiding significance to our researches. We have studied comments carefully and have made correction which we hope meet with approval. And the revised portion is marked in red.

1.The comment: However, there are many points in the paper that need to be modified and improved, such as the manufacturers and batch numbers of the main instruments provided for research;

The answer: Thank you very much for your sincere suggestion. We have added the manufacturers and batch numbers of the main instruments provided for research in the manuscript.

2.The comment: Provide the strain, specification and qualification certificate number of mice used in the test; The licence number of the animal centre for the supply of mice.

The answer: Thank you for the suggestion. The mice was SPF BALB/c male mice, weighing 18-20 g, the qualification certificate number of mice is SYXK (Chuan)2020-196. And the licence number of the animal centre for the supply of mice is SCXK(Chuan)2020-0030. And it's been added to the manuscript.

3.The comment: There should be references in the method.

The answer: The references cited have been added to the methods section.

4.The comment:There should be model group, positive control group, and the best three dose groups in the animal experiment.

The answer: Thank you very much for your sincere suggestion. Adding a positive control group does help to improve the quality of the paper. Unfortunately, we did not consider adding positive drugs in this study. The purpose of this study was to preliminarily investigate the in vivo inhibitory effect of ISL on colorectal tumors. Therefore, just the high-dose and low-dose groups were included in this study and compared with the model group to calculate tumor inhibition rate. And in similar studies, there was no positive control in examining the anti-tumor effects of the compound (Song et al., 2020; Lv et al., 2023; Zhao et al.,2023 ). Indeed, appropriate positive drug controls and more dose groups should be considered in our future studies.

Song L, Luo Y, Li S, Hong M, Wang Q, Chi X, Yang C. ISL Induces Apoptosis and Autophagy in Hepatocellular Carcinoma via Downregulation of PI3K/AKT/mTOR Pathway in vivo and in vitro. Drug Des Devel Ther. 2020 Oct 20;14:4363-4376. 

Lv C, Huang Y, Wang Q, Wang C, Hu H, Zhang H, Lu D, Jiang H, Shen R, Zhang W, Liu S. Ainsliadimer A induces ROS-mediated apoptosis in colorectal cancer cells via directly targeting peroxiredoxin 1 and 2. Cell Chem Biol. 2023 Mar 16;30(3):295-307.e5.

Zhao P, Song H, Gao F, Chen L, Qiu J, Jin J, Pan C, Tang Y, Chen M, Pan Y, Li Y, Huang L, Yang J, Hao X. A Novel Derivative of Curcumol, HCL-23, Inhibits the Malignant Phenotype of Triple-Negative Breast Cancer and Induces Apoptosis and HO-1-Dependent Ferroptosis. Molecules. 2023 Apr 12;28(8):3389.

5.The comment: How to make mold should have a detailed method and mold detection index;

The answer: The details of the model making and detection index have been added to the methods section for Animals and Treatment.

6.The comment: The content of animal experiments in this study is small, with 6 mice in each group, which is a little small.

The answer: Considering the welfare ethics of experimental animals, 6 mice in each group were selected for animal experiments to meet the minimum sample size required for statistical purposes. In similar studies, 5-6 mice per group were recognized (Wang et al., 2023; Zhang et ai., 2023; Li et al., 2023).

Wang YH, Wang QT, Wu XT, Dong Y, Cui Q, Zhou YN, Yang XW, Lu WF, Li WY, Wang H, Zhao XD, Zhang M. Yiqi Yangjing recipe stimulates apoptosis while suppressing the energy metabolism via under-expression of PFKFB3 in A549 cells. J Thorac Dis. 2023 Sep 28;15(9):4885-4895. doi: 10.21037/jtd-23-490.

Zhang X, Huang J, Wang J, Li Y, Hu G, Li H. Circ_0001667 accelerates breast cancer proliferation and angiogenesis through regulating CXCL10 expression by sponging miR-6838-5p. Thorac Cancer. 2023 Apr;14(10):881-892.

Li F, Song X, Zhou X, Chen L, Zheng J. Emodin attenuates high lipid-induced liver metastasis through the AKT and ERK pathways in vitro in breast cancer cells and in a mouse xenograft model. Heliyon. 2023 Jun 7;9(6):e17052.

7.The comment: The discussion part should discuss what this study reveals, the existing problems and further research in the future. Some content can be appropriately added to the conclusion.

The answer:Thanks for your suggestion, those have been added to the conclusion.

8.The comment: The references should be mainly from the last 3 years, and the format should be uniform. In view of the above problems, it is suggested to revise and improve the trial.

The answer: Thanks very much for your helpful comments. We have revised the references mainly from the last 3 years.

Thank you very much for your attention and time, look forward to hearing from you!

Yours sincerely,

Feng-lin Luo

2, Dec, 2023

Reviewer 3 Report

Comments and Suggestions for Authors

This article presents the results of a study on the effect of ISL against CRC and its possible mechanism. The article may be of interest to specialists working in the field of research into treatment methods for colorectal cancer.

A large number of typos were found in the article, some of them are given below: immunotherapy[4–9] - a space is required throughout the text; in vivo - it is necessary to make “italics” (page 2, 15); GO, TNF, etc. - it is necessary to decipher abbreviations the first time they are mentioned in the text; CO2 - typo (page 13); In total, 1×107 CT-26 cells were implanted; 40-60mm3 - typos (page 14).

Also, please check the text for other typos.

In addition, it is necessary to expand the “conclusion” section and add more conclusions about the results obtained.

After correcting all typos, the article may be revised.

Comments on the Quality of English Language

Moderate editing of English language required.

Author Response

Dear reviewer,

Thanks for your comments concerning our manuscript entitled “Isoliquiritigenin inhibits the growth of colorectal cancer cells through the ESR2/PI3K/AKT signalling pathway” (Manuscript ID: pharmaceuticals-2702681). Those comments are all valuable and very helpful for revising and improving our paper, as well as the important guiding significance to our researches. We have studied comments carefully and have made correction which we hope meet with approval. And the revised portion is marked in red bold.

1.The comment: A large number of typos were found in the article, some of them are given below: immunotherapy[4–9] - a space is required throughout the text; in vivo - it is necessary to make “italics” (page 2, 15); GO, TNF, etc. - it is necessary to decipher abbreviations the first time they are mentioned in the text; CO2 - typo (page 13); In total, 1×107 CT-26 cells were implanted; 40-60mm3 - typos (page 14).

Also, please check the text for other typos.

The answer: Thanks for your sincere reminding. We apologize for the mistakes. The typos in the article have been revised.

2.The comment: In addition, it is necessary to expand the “conclusion” section and add more conclusions about the results obtained. 

The answer: Thank you very much for your sincere suggestion. We have expanded the conclusion in the revised manuscript.

3.The comment: Moderate editing of English language required.

The answer: We've submitted the manuscript to the English editing service provided by MDPI for revision (English editing ID: english-73052).

Thank you very much for your attention and time, look forward to hearing from you!

Yours sincerely,

Feng-lin Luo

4, Dec, 2023

Reviewer 4 Report

Comments and Suggestions for Authors

-In section 4.2, Molecular Docking:

Please provide more details, such as the database you used to extract the structure of the targets and ligands. Additionally, include the PDB code of the targets if you used the PDB database. Also, explain how you optimized your ligand.

-Furthermore, please add 2D schematics of interactions to Figure 2.

Author Response

Dear reviewer,

Thanks for your comments concerning our manuscript entitled “Isoliquiritigenin inhibits the growth of colorectal cancer cells through the ESR2/PI3K/AKT signalling pathway” (Manuscript ID: pharmaceuticals-2702681). Those comments are all valuable and very helpful for revising and improving our paper, as well as the important guiding significance to our researches. We have studied comments carefully and have made correction which we hope meet with approval. And the revised portion is marked in red.

1.The comment: In section 4.2, Molecular Docking: Please provide more details, such as the database you used to extract the structure of the targets and ligands. Additionally, include the PDB code of the targets if you used the PDB database. Also, explain how you optimized your ligand. 

The answer: The database we used to extract the structure of the targets and ligands was PDB database ( https://www.rcsb.org/ ). The PDB codes for ESR2, PI3KCG and GSK3β proteins are 7xvz, 4g11 and 5f94, respectively (please see “Supplementary Materials for molecular docking” for details ). In order to optimize the ligand, Firstly, we used pymol software to split the protein and ligand, then removed the water molecules of the protein as well as the redundant amino acid sequences. Finally, we extracted the original ligand, and exported the processed protein and the original ligand, and saved them in PDB format. And those details have been added to the article.

2.The comment:Furthermore, please add 2D schematics of interactions to Figure 2.

The answer: Thanks for your suggestion. We have added 2D schematics of interactions to Figure 2.

Thank you very much for your attention and time, look forward to hearing from you!

Yours sincerely,

Feng-lin Luo

4, Dec, 2023

Reviewer 5 Report

Comments and Suggestions for Authors

The authors investigated the potential molecular mechanism of Isoliquiritigenin against CRC.

Page.2- When the authors begin describing Isoliquiritigenin (2 ',4 ',4-trihydroxy chalcone; ISL) for the first time, they should also highlight few other uses of the same compound.

A new discipline called network pharmacology (NP) is new concept to many readers. It should be explained in the introduction section.

NP has the potential to provide new treatments to multigenic complex diseases and can lead to the development of e-therapeutics. Please include these facts.

Tumor necrosis factor (TNF)-related apoptosis-inducing ligand (TRAIL) and colon cancer could be elaborated. Combination of isoliquiritigenin and TRAIL induces nuclear fragmentation. isoliquiritigenin regulates proteins which relate to TRAIL signaling and enhances TRAIL-induced apoptosis. These facts could be discussed.

What is the optimal concentrations of isoliquiritigenin?

 Isoliquiritigenin up-regulates death receptor 5 (DR5) expression. This is important in cancer.

Can Isoliquiritigenin inhibit the viability of cancer cells in a concentration- and time-dependent manner?

ISL reportedly inhibits the growth of different cancer cells in which phase of the cell cycle?

What is the role of autophagy in this cancer?

Autophagy inhibitors are classified as early- or late-stage drugs according to their mechanisms of action on the pathway inducing autophagy. Which one is important?

Few limitations could be stated.

Latest references from 2022-2024 should be included.

Author Response

Dear reviewer,

Thanks for your comments concerning our manuscript entitled “Isoliquiritigenin inhibits the growth of colorectal cancer cells through the ESR2/PI3K/AKT signalling pathway” (Manuscript ID: pharmaceuticals-2702681). Those comments are all valuable and very helpful for revising and improving our paper, as well as the important guiding significance to our researches. We have studied comments carefully and have made correction which we hope meet with approval. And the revised portion is marked in red.

1. The comment:Page.2- When the authors begin describing Isoliquiritigenin (2 ',4 ',4-trihydroxy chalcone; ISL) for the first time, they should also highlight few other uses of the same compound.

The answer: Thanks for your helpful suggestion. The other pharmacological effects of ISL have been added in the introduction.

2.The comment: A new discipline called network pharmacology (NP) is new concept to many readers. It should be explained in the introduction section. NP has the potential to provide new treatments to multigenic complex diseases and can lead to the development of e-therapeutics. Please include these facts.

The answer:Thanks for your suggestion. We have added the description of network pharmacology in the introduction.

3.The comment: Tumor necrosis factor (TNF)-related apoptosis-inducing ligand (TRAIL) and colon cancer could be elaborated. Combination of isoliquiritigenin and TRAIL induces nuclear fragmentation. isoliquiritigenin regulates proteins which relate to TRAIL signaling and enhances TRAIL-induced apoptosis. These facts could be discussed.

The answer:Thank you for the suggestion.Those have been added to the discussion.

4.The comment: What is the optimal concentrations of isoliquiritigenin?

The answer: In this study, ISL could concentration-dependently inhibit the activity of colorectal cancer HCT116 and SW480 cells with IC50 values of 53.48 and 101.7 μmol/L, respectively.

5.The comment: Isoliquiritigenin up-regulates death receptor 5 (DR5) expression. This is important in cancer.

The answer:Thank you very much for your sincere reminder. We searched and read the relevant reports on the regulation of ISL on DR5. And we understand the importance of DR5 regulation in the treatment of colorectal cancer, and will consider it in future studies.

6.The comment: Can Isoliquiritigenin inhibit the viability of cancer cells in a concentration- and time-dependent manner?

The answer: Yes, Isoliquiritigenin can inhibit the viability of cancer cells in a concentration- and time-dependent manner. In this study, the viability of colorectal cancer cells HCT116 and SW480 gradually decreased with increasing concentrations of ISL. Some previous studies have also shown that Isoliquiritigenin can inhibit tumor cell growth in a concentration-dependent and time-dependent manner. The references are as follows:

Tian T, Sun J, Wang J, Liu Y, Liu H. Isoliquiritigenin inhibits cell proliferation and migration through the PI3K/AKT signaling pathway in A549 lung cancer cells. Oncol Lett. 2018 Nov;16(5):6133-6139. doi: 10.3892/ol.2018.9344. 

Peng F, Tang H, Liu P, Shen J, Guan X, Xie X, Gao J, Xiong L, Jia L, Chen J, Peng C. Isoliquiritigenin modulates miR-374a/PTEN/Akt axis to suppress breast cancer tumorigenesis and metastasis. Sci Rep. 2017 Aug 21;7(1):9022. doi: 10.1038/s41598-017-08422-y.

Zhang B, Lai Y, Li Y, Shu N, Wang Z, Wang Y, Li Y, Chen Z. Antineoplastic activity of isoliquiritigenin, a chalcone compound, in androgen-independent human prostate cancer cells linked to G2/M cell cycle arrest and cell apoptosis. Eur J Pharmacol. 2018 Feb 15;821:57-67. doi: 10.1016/j.ejphar.2017.12.053.

7.The comment: ISL reportedly inhibits the growth of different cancer cells in which phase of the cell cycle?

The answer: Lin et al. reported that ISL reduced Triple-Negative Breast cell cycle progression through the reduction of cyclin D1 protein expression and increased the sub-G1 phase population. Zhang et al. found that ISL (25-50μM) inhibited the proliferation of  prostate cancer cell, induced cell apoptosis, and caused G2/M cell cycle arrest in vitro. Chen et al. showed that ISL significantly inhibited the viability of ovarian cancer cells in a concentration- and time-dependent manner and induced G2/M phase arrest. In addition, our study also showed that ISL can inhibit the proliferation of CRC cells and block the cell cycle in the G2/M phase. These findings suggest that ISL may inhibit cancer cell growth in the G2/M or sub-G1 phase of the cell cycle. The references are as follows:

Lin PH, Chiang YF, Shieh TM, Chen HY, Shih CK, Wang TH, Wang KL, Huang TC, Hong YH, Li SC, Hsia SM. Dietary Compound Isoliquiritigenin, an Antioxidant from Licorice, Suppresses Triple-Negative Breast Tumor Growth via Apoptotic Death Program Activation in Cell and Xenograft Animal Models. Antioxidants (Basel). 2020 Mar 10;9(3):228. doi: 10.3390/antiox9030228.

Zhang B, Lai Y, Li Y, Shu N, Wang Z, Wang Y, Li Y, Chen Z. Antineoplastic activity of isoliquiritigenin, a chalcone compound, in androgen-independent human prostate cancer cells linked to G2/M cell cycle arrest and cell apoptosis. Eur J Pharmacol. 2018 Feb 15;821:57-67. doi: 10.1016/j.ejphar.

Chen HY, Huang TC, Shieh TM, Wu CH, Lin LC, Hsia SM. Isoliquiritigenin Induces Autophagy and Inhibits Ovarian Cancer Cell Growth. Int J Mol Sci. 2017 Sep 21;18(10):2025. doi: 10.3390/ijms18102025.

8.The comment: What is the role of autophagy in this cancer?

The answer: Autophagy is a physiological process by which intracellular components are degraded in their own lysosomes. In recent years, targeting autophagy for cancer treatment has received increasing attention. Studies have shown that autophagy has a dual role in the development of colorectal cancer. On the one hand, autophagy activation can inhibit the development of colorectal cancer and induce autophagic death of colorectal cancer cells. On the other hand, autophagy can also maintain the cellular activity of colorectal cancer in hostile environments (e.g. starvation).The references are as follows:

Amaravadi RK, Kimmelman AC, Debnath J. Targeting Autophagy in Cancer: Recent Advances and Future Directions. Cancer Discov. 2019 Sep;9(9):1167-1181. doi: 10.1158/2159-8290.CD-19-0292.

Li S, Wang X, Wang G, Shi P, Lin S, Xu D, Chen B, Liu A, Huang L, Lin X, Yao H. Ethyl Acetate Extract of Selaginella doederleinii Hieron Induces Cell Autophagic Death and Apoptosis in Colorectal Cancer via PI3K-Akt-mTOR and AMPKα-Signaling Pathways. Front Pharmacol. 2020 Oct 19;11:565090. doi: 10.3389/fphar.2020.565090.

8.The comment: Autophagy inhibitors are classified as early- or late-stage drugs according to their mechanisms of action on the pathway inducing autophagy. Which one is important? Few limitations could be stated.

The answer: Autophagy inhibitors are categorized as the early-stage inhibitors targeting ULK1/ULK2 or VPS34, and the late-stage inhibitors targeting the lysosome.

The early-stage autophagy inhibitors mainly include pan-PI3K inhibitors, Vps34 inhibitors and ULK inhibitors. These compounds inhibit the growth of tumor both in vitro and in vivo in various cancer types via dual autophagy. They can significantly improves sensitivity to chemotherapeutic drugs, enhance the efficacy of anti-programmed death-ligand 1 (PD-L1)/programmed cell death-1 (PD-1) blockade, etc. However, the specificity, tolerability and a range of side effects such as muscle toxicity limit their use to laboratory tools.

The late-stage autophagy inhibitors also show dual character in cancer treatment. For exmple, CQ and HCQ inhibit autophagosome degradation by interfering with lysosomal acidification. However, both drugs then provoke lysosomal compartment deacidification, stopping lysosomal enzyme activity, resulting in inhibition of autophagy flux and disruption of autophagosome-lysosome fusion. There are a number of the unexpected adverse effects of CQ treatment, such as severe constipation, gastrointestinal toxicity, cardiomyopathy, grade 3 fatigue, anemia, or retinal toxicity, which have been monitored in various cancer types.

Unfortunately, We did not investigate interventions on autophagy in the treatment of colorectal cancer in our previous studies. And the effect of ISL on autophagy will be considered in subsequent studies.

Pasquier B. Autophagy inhibitors. Cell Mol Life Sci. 2016 Mar;73(5):985-1001.

Onorati AV, Dyczynski M, Ojha R, Amaravadi RK. Targeting autophagy in cancer. Cancer. 2018 Aug;124(16):3307-3318.

Mauthe, M.; Orhon, I.; Rocchi, C.; Zhou, X.; Luhr, M.; Hijlkema, K.-J.; Coppes, R.P.; Engedal, N.; Mari, M.; Reggiori, F. Chloroquine Inhibits Autophagic Flux by Decreasing Autophagosome-Lysosome Fusion. Autophagy 2018, 14, 1435–1455.

Bestion E, Raymond E, Mezouar S, Halfon P. Update on Autophagy Inhibitors in Cancer: Opening up to a Therapeutic Combination with Immune Checkpoint Inhibitors. Cells. 2023 Jun 23;12(13):1702.

8.The comment: Latest references from 2022-2024 should be included.

The answer: Thanks very much for your helpful comments. We have revised the references mainly from the last 3 years.

Thank you very much for your attention and time, look forward to hearing from you!

Yours sincerely,

Feng-lin Luo

8, Dec, 2023

Round 2

Reviewer 3 Report

Comments and Suggestions for Authors

I thank the authors for their attentive attitude to the reviewer’s comments.

Comments on the Quality of English Language

Minor editing of English language required